# Regulation of Bone by Mechanical Loading, Sex Hormones, and Nerves: Integration of Such Regulatory Complexity and Implications for Bone Loss during Space Flight and Post-Menopausal Osteoporosis

**DOI:** 10.3390/biom13071136

**Published:** 2023-07-15

**Authors:** David A. Hart

**Affiliations:** Department of Surgery, Faculty of Kinesiology, and McCaig Institute for Bone & Joint Research, University of Calgary, Calgary, AB T2N 4N1, Canada; hartd@ucalgary.ca

**Keywords:** bone regulation, mechanical loading, sex hormones, neural regulation, fracture healing, spinal cord injury, space flight, menopause, brain trauma

## Abstract

During evolution, the development of bone was critical for many species to thrive and function in the boundary conditions of Earth. Furthermore, bone also became a storehouse for calcium that could be mobilized for reproductive purposes in mammals and other species. The critical nature of bone for both function and reproductive needs during evolution in the context of the boundary conditions of Earth has led to complex regulatory mechanisms that require integration for optimization of this tissue across the lifespan. Three important regulatory variables include mechanical loading, sex hormones, and innervation/neuroregulation. The importance of mechanical loading has been the target of much research as bone appears to subscribe to the “use it or lose it” paradigm. Furthermore, because of the importance of post-menopausal osteoporosis in the risk for fractures and loss of function, this aspect of bone regulation has also focused research on sex differences in bone regulation. The advent of space flight and exposure to microgravity has also led to renewed interest in this unique environment, which could not have been anticipated by evolution, to expose new insights into bone regulation. Finally, a body of evidence has also emerged indicating that the neuroregulation of bone is also central to maintaining function. However, there is still more that is needed to understand regarding how such variables are integrated across the lifespan to maintain function, particularly in a species that walks upright. This review will attempt to discuss these regulatory elements for bone integrity and propose how further study is needed to delineate the details to better understand how to improve treatments for those at risk for loss of bone integrity, such as in the post-menopausal state or during prolonged space flight.

## 1. Introduction

### 1.1. Purpose of This Review

As life evolved on Earth, various systems adapted to living within the boundary conditions of the planet. These boundary conditions include the 1 g gravity and the geomagnetic field (GMF). As life forms became more complex, they required a skeletal framework to stabilize the various organs and tissues. For humanoids who walk upright, this also required additional modifications to the skeleton to accommodate the added height and need to maintain the integrity of the blood supply to areas of the body, such as the brain, an organ using a considerable effort in this regard. The integrity of the skeleton and the associated tissues allowing for mobility, such as muscles, tendons, and ligaments, became central to survival. Mobility also required the development of joints, such as the ankle, knee, and hip, as well as a multisegmented spine. All of these tissues subscribe to the “use it or lose it” paradigm (discussed in [1,2]) in that lack of use leads to atrophy of the tissues [3,4,5]. This concept has led to the further understanding that tissues, such as bone in the skeleton, are regulated primarily by mechanical considerations, with a focus on the bone itself as the central consideration. However, three different circumstances shed some doubt on the concept that mechanical loading is the only regulator of bone integrity. The first is post-menopausal osteoporosis, which develops in a subset of females after menopause. The second is the response of bone to space flight and microgravity, a condition that could not have been predicted by evolution. Bone loss in microgravity does not appear to be completely prevented or reversed by exercise and loading. The third is bone loss following a spinal cord injury or hemiplegia after a stroke. This review will discuss those situations in some detail and attempt to integrate the discussion with the literature regarding the mechanical loading of bone being the primary stimulus for maintaining the integrity of the tissue in different loading environments with additional consideration for regulation by sex hormones and neural elements.

### 1.2. Regulation of Bone across the Lifespan

When humans are born, they are not mobile and require a year or more until they are able to walk, and then they usually walk in a straight line, only later learning to navigate their environment. In contrast, animals, such as wildebeests or zebras, are born and then are up and running within hours. Obviously, humans are protected from harm by their mothers and family, while for the indicated animals, early mobility is critical for survival. However, it is curious that humans take so long to gain upright mobility and the ability to navigate their environment after a nine-month gestation.

From the above, apparently humans require some growth and maturation within the boundary conditions of Earth before they have the skeletal ability and neurological mechanisms sufficiently matured to be able to accomplish an upright stance, walking, and environmental navigation. While some of this maturation before upright walking and then navigation may be due to the complexities of the integration of neural regulation into coherent mobility, it may just be a stochastic compromise between gestation time and function for other reasons. One other reason may be genetic, as some children in the same family start walking at 7–8 months of age, while others do not walk until 12–14 months of age. While boundary conditions of Earth, such as the 1 g gravity do vary somewhat in different locations [6], no association between gravitational field and time to walking could be found. Similarly, the GMF does vary with location, as does the background magnetic fields due to local concentrations of iron containing minerals vary (discussed in [6]). However, again, no references relating local and GMF strengths to time to walking and navigation could be found. Therefore, either these two boundary conditions do not play a role, or it has not been investigated and published.

During the period between birth and walking, the skeletal system grows significantly with bone and muscle growth, which coordinate growth between the arms and legs. While not being subjected to ground reaction force (GRF) loading, the muscles and joints are developing via motion along with the bones. How motion and loading are required for the integrated progression during growth of a motion segment, such as a leg, was shown by preclinical model studies in very young rabbits [7,8]. If one leg of a very young rabbit (4 weeks of age) was rigidly immobilized in flexion, the knee ligaments ceased to grow, but the bones of the lower leg continued to grow such that the knee could not be straightened when the pin was removed after 6–8 weeks. Interestingly, the left and right legs were very similar in length. If the pinning was delayed until ~8 weeks of age, such overgrowth of the bones of the lower leg did not occur. Thus, in this model, the bones could continue to grow via the growth plates in the absence of movement or non-GRF loading very early, but this dysregulation slowed down with time after birth. Whether this early bone growth was due to the presence of an “excess” of anabolic factors or neural influences, which gradually declined in this model, or some other variable remains undefined. Similarly, it is not known whether this type of bone growth occurs in humans in the post-natal time frame. However, as humans uniquely walk upright, this lengthy period from birth to walking may provide the needed bone and muscle coordination to assume this upright position for walking prior to subjecting the components to GRF.

Aside from time to walking and pre-walking events, two other phases of life are critical for skeletal maturation. These are growth and maturation prior to puberty (ages ~1–puberty) and then puberty to skeletal maturity (~11–13 to ~19–21 years of age). Prior to puberty, growth in males and females is fairly similar, with variation in details likely due to genetics. This growth likely is dependent on engaging the GRF while walking and running, leading to the concept that mechanical loading is required to work with the anabolic stimuli leading to bone growth and density appropriate for the strains and stresses resulting from the GRF. Thus, this is a very dynamic period in the lifespan as tissues, such as bones and muscles that collaborate to form “muscle–bone units” [9,10,11,12]. Furthermore, activated muscle and bones release myokines and osteokines, respectively, which can affect many biological tissues (discussed in [1,5]). As bones and muscles both subscribe to the “use it or lose it” principle unless actively engaged, they can atrophy (discussed in [1,5]).

With the onset of puberty, not only is there significant growth but also development of many sex differences in the musculoskeletal system, including bone [13]. For females, there can be a widening of the hips, the onset of spinal conditions, such as adolescent idiopathic scoliosis [14,15], as well as increased muscle and bone changes. For males, the increase in muscle mass with the onset of puberty may be due to testosterone, but in females, the bone adaptations may be more estrogen-dependent [13]. Interestingly, prior to puberty, males and females jump from a height and land very similarly, but after puberty girls land differently than boys [16,17,18], and some of these differences may be influenced by strength training [19]. Thus, puberty leads to many alterations, including to bone and muscle and their regulation.

Accompanying puberty in females are a number of menstrual cycle-associated and hormone-related events regarding the musculoskeletal system (MSK) system. These include changes to knee joint laxity in most, but not all females [20,21], risk for developing an imbalance between the hamstring and quadriceps muscles [22,23], and menstrual cycle-dependent bone turnover [24,25,26]. In males, puberty-associated increases in testosterone have been implicated in muscle changes but not bone changes [13]. After skeletal maturity, pregnancy and lactation are also associated with bone turnover [27,28], much of which can apparently be reversed over time [29]. Rodents can lose 25–35% of their bone calcium during pregnancy and lactation, leading to compromised bone strength (reviewed in [29]). Such bone loss is not uniform, and the calcium loss during lactation is mainly from the spine (reviewed in [29]). In humans, bone loss and calcium mobilization may be 5–10% of the reservoir in the spine. Post-lactation, the bone integrity can recover and replace that lost via lactation by mainly unknown mechanisms. In rare cases, some females can develop pregnancy and lactation osteoporosis that may not recover readily [30]. Whether the bone that is lost during pregnancy and lactation is a unique subset of bone is not well defined, but the tissue-specific aspects would lend some credibility to the concept that not all bone is affected equally.

Finally, during aging and senescence in females, a subset develops osteoporosis (OP), with variable rates of loss of bone integrity (reviewed in [31]). Why only a subset of females develop OP is not really known in detail. Given the variable rate of bone loss and the variable sites of the bone loss [32,33], the disease may have at least two components, one that initiates the disease and the other that is involved in the regulation of the rate of bone loss. The majority of bone is lost from the lumbar spine in many females [33], not unlike what has been observed during pregnancy and lactation [29]. While ~75% of OP cases are female, a subset of males also appears to develop an age-related form of the disease. The basis for why this subset is affected is largely unknown. Interestingly, both females and males with OP can respond to anti-resorptive drugs, such as bisphosphonates and monoclonal antibody-type reagents with specificity to other relevant molecules (discussed in [34]). Thus, after menopause in this subset of females that experience OP, there is bone loss again in the spine and to a lessor extent the forearm and the hip [35,36,37]. Whether there is a subset of bone that is affected early after onset of menopause is not well defined, but certainly such a concept is supported by the studies reported by Shieh et al. [33]. While such information may be of interest to better understanding the mechanisms associated with post-menopausal bone loss and that associated with pregnancy and lactation, the main issue to be elucidated is how bone recovery is accomplished following pregnancy and lactation but appears to be an open-ended continual decline in bone integrity following menopause in females, and how it may differ in age-related OP in a subset of males.

Therefore, bone is regulated differently across the lifespan and is also regulated differently in females and males. Furthermore, such regulation is also variable in different females and males as the onset of puberty, menopause, and aging/senescence affects different individuals differently, and thus, heterogeneity exists in the regulation of bone and other MSK components. Thus, genetic heterogeneity between individuals likely also contributes to the complexity of the regulation of bone and the interrelationship between mechanical loading, sex hormones, and neural influences.

## 2. The Regulation of Bone by Mechanical Loading

As mentioned above, bone appears to subscribe to the “use it or lose it” paradigm, with an ability to adapt to variation in loading environments (discussed in [1,2,5]). This latter adaptation to the stress needs within a tissue also means that within a bone, such as the femur, the set point for bone density may be dictated by the stress environment (discussed in [3,38]) and Wolff’s law (discussed in [4]). However, if the adaptation to a changing stress environment does not proceed at a rate that can be handled, allowing for time to alter cellular activities, the tissue can fatigue, and integrity compromised. While much research activity has focused on osteoblasts (OB) and osteoclasts (OC) in the regulation of bone, it is clear that the actual mechanical sensing cells may be the osteocytes [39] playing a regulatory role in bone [40,41], potentially using cellular products in the cells from the piezo family of genes as sensors (discussed in [42,43,44]). Other cellular components, such as the parathyroid hormone receptor type 1 may also be involved in mechanosensing by bone cells [45,46,47]. However, as the number and location of osteocytes is somewhat heterogeneous (discussed in [48]) and likely regulated by intraosseous fluid flow [49,50], mechano-sensing osteocytes would have to have an effective mechanism to communicate with OB and OC in order to translate mechanical signals to alterations in the activity of the OB and OC. One possible mechanism could be the release of unique cytokines that function in a paracrine manner (discussed in [51]).

While the situation in vivo regarding cell–cell communication and regulation of bone structure and density is likely complex, it is clear that bone cells, such as the osteoblastic cell line MG-63, can respond to mechanical loading in vitro [52] and that explants of bone can also respond to mechanical loading in vitro [53]. Therefore, these cells can respond to loading independent of other in vivo regulatory factors.

However, loading of bones of the lower extremities via ground reaction forces (GRF) must go through some discontinuities at the ankle, knee, and hip to reach the bones of the vertebral column. In particular, the knee joint not only is discontinuous (between the tibia and femur), but there is also synovial fluid (SF) in the knee joint itself that could dissipate the loading. While some of such loading may lead to direct surface-to-surface contact, the surfaces of the bone are covered by articular cartilage that is mainly water. Therefore, it is unclear how loading at the foot/heel leads to mechanical forces being conveyed to the hip and beyond if such discontinuities disrupt the transmission of the loading along the lower extremity motion segment.

Alternatively, as the ankle and heel bone are innervated [54,55], and one can sense feeling from the foot to the brain, nerves traverse the whole length of the lower limb and meet at the dorsal root ganglion (discussed in [56,57,58]). Thus, if loading at the foot led to nerves receiving information following mechanical stimulation, such information could be transmitted past the discontinuities and allow for the information to also be received back to the femur and vertebral column. Of note in this regard are the report that a subset of neurons in connective tissues other than bone express members of the piezo gene family and such cells may be involved in mechanical sensing [59]. Thus, there is a precedent for neural elements to be able to respond to mechanical loading but whether bone also contains such piezo positive neural cells remains to be determined. Whether relevant information from mechanical stimulation of neural elements in bone could be transmitted to the whole bone or just specific areas must also await future research.

Such a system might explain the loss of bone being regulated differently than muscle in space. With the loss of gravity and GRF sensing in microgravity, there could be a differential loss of the neural regulation that is not replicated by the mainly resistive exercises performed in space. Thus, loss of impact loading at the heel in space may lead to preferential loss of the neural input into bone regulation. This concept, admittedly a hypothesis at this point, could be tested by developing an impact-loading device for assessing influence in space environments on the International Space Station (ISS).

The involvement of such a neuroregulatory mechanosensitive system for bone regulation might also be influenced by factors on Earth as well and contribute to bone regulation. That is, during evolution, humans and their predecessors likely went barefoot for much of the time on land. Thus, in the relatively more recent time, humans have taken to wearing foot coverings, such as sandals, shoes, boots, and running shoes, designed to reduce the wear and tear on the foot and protect it from GRF! The advantages and disadvantages of running barefoot has been discussed by Nigg and Enders [60], as well as the assessment of barefoot walking gait kinetics by post-menopausal females [61]. However, comparisons between barefoot walking/running versus shod movement on bone density could not be found in the available literature. While some barefoot runners may land using the forefoot or the heel, walkers usually have a heel strike first so future studies could assess whether wearing shoes or other “protective” footwear alter the impact of walking on bone density. Perhaps shod feet are negating an evolutionary enabler of bone density in order to protect feet from hard surfaces, such as concrete, asphalt, and wood or to protect them from sharp objects!

## 3. Neural Regulation of Bone

### 3.1. Background

That bone is innervated has been known for a long time (discussed in [62,63,64,65]), and the human femur has been shown to have considerable innervation associated with cortical canals [66]. This innervation of bone has been implicated in skeletal development, adaptation, and aging [67,68], as well as healing of the tissue [69]. Furthermore, crosstalk between bone/the skeletal system and neural tissues may be important for maintaining skeletal health via a neuro-osteogenic network (reviewed in [70,71]). This network could be manifested by mediators released from central nervous system centers or the local release of neuropeptides from the innervation of the bone directly. Recently, Liu et al. [72] have reviewed the possibility of the “quaternary regulation” theory of a “peripheral nerve–angiogenic–osteoclast–osteogenesis” network in a network mediated by neuropeptides. Thus, neuropeptides, such as neuropeptide Y (NPY), substance P, or calcitonin gene-related peptide (CGRP) released locally [73,74,75], could influence the metabolic activity of bone cells (i.e., osteoblasts, osteoclasts, osteocytes) [76,77,78,79]. Neuropeptides, such as CGRP, may have a role to play in bone remodeling [80], in the treatment of osteoporosis [81] or in fracture healing [82]. Such neuro regulation could occur in collaboration with mechanical loading [83] or energy utilization during different stages of growth and aging [84].

Regarding aging, it has been shown that while innervation is likely important in the growth of long bones, there can also be a loss of innervation with aging. This has been mainly investigated in rodent models [68,85,86,87]. Loss of innervation with aging could lead to compromised bone regulation and contribute to the development of bone diseases, such as OP. However, this will require additional confirmation via research. Of note, loss of knee joint innervation with aging in C57/BL6 mice has shown a relationship with the development of osteoarthritis [88,89]. In rodents, such afferents contain substance P and CGRP [85]. Therefore, innervation is not static across the lifespan, loss in bone is likely not unique, and such loss can also occur in other tissues during the aging process.

### 3.2. Loss of Neural Integrity on Bone

Bone loss after the transection of nerves to induce denervation has been shown in a variety of rodent models [90,91,92,93,94]. These model systems mainly attributed the bone loss to disuse. While an “artificial” system, these models did provide some insights. Brouwers et al. [91] reported that, as assessed by microCT approaches, the bone loss after denervation differed from that induced by surgical menopause in rats. Ma et al. [92] reported that naringin administration could ameliorate bone loss after surgical denervation in rats. Naringen is a flavonoid with anti-inflammatory actions in rats [95]. Interestingly, use of electrical [96,97] or electromagnetic fields [94] were also shown to decrease bone loss or restore bone after denervation. As surgical denervation models also lead to muscle atrophy and loss of the muscle–bone unit [98], the effects of denervation on bone integrity could be indirect or direct consequences of the surgical denervation on the bone. Deng et al. [99] have reported that surgical denervation compared to use of botox to induce muscle paralysis has differing effects on bone activity, so denervation may actually have both direct and indirect impact on bone responses.

Further evidence for neural regulation of bone has come from the literature regarding bone changes occurring after disruption of the neural regulation induction of “neurogenic osteoporosis” [100]. Two of the more well studied circumstances in patients has been following a spinal cord injury [101,102,103] and following a hemiplegic stroke [104,105].

Following a stroke, there are skeletal consequences that result in altered bone regulation and development of osteoporosis with increased risk for fractures [106,107]. There can be rapid bone loss, particularly on the side with plegia, and the overall pattern of bone loss is somewhat different from that arising in the post-menopause state (reviewed in [104,108]). The causes of the bone loss are not well characterized but may be due to inactivity, vitamin D deficiency, or secondary effects of the stroke [104,108], but direct effects of the loss of neural regulation have not been discussed in any detail. Interestingly, attempts to prevent or reverse the post-stroke bone loss using drug interventions (discussed in [108]), as well as non-pharmacological interventions, such as exercise [109,110,111], have been reported. These approaches have been partially effective, but some localized responses have been noted and drug variation in responsiveness.

A second condition that also supports a role for nerves in bone health and responses to stimuli is the development of osteoporosis in spinal cord injury patients (SCI) [101,112,113,114,115,116]. Loss of bone is restricted to bones that are below the level of the injury [116,117], and the bone loss can occur rapidly and for a prolonged period of time. However, the bone loss can be site- or location-specific and vary between patients [113,118,119]. The bone loss associated with a spinal cord injury can be partially prevented by treatment with bisphosphonates [117,120,121]. However, treatment with such drugs exhibits some site-specific responses in that they are effective in preserving bone at the hip, but much less so at the knee [117]. Other approaches reporting some success in preventing bone loss with SCI include activity-based physical therapy [122] and functional electrical stimulation [123].

As the knee is a more frequent site for fractures than the hip, treatment with bisphosphonates may not prevent bone loss in a strategic manner. Furthermore, such findings may indicate that the knee is a unique site when it comes to neuroregulation. Previously, Hart [124] proposed that the knee was involved in a unique “knee–eye–brain axis” that was intimately involved in regulating mobility and navigation. Disruption of such an axis by knee injuries was the initial focus, but certainly disruption of the integrity of such an axis by a spinal cord injury would lead to loss of neural output from the knee as well as input from the brain and eye to the knee as well. The latter could also contribute to tissue integrity, including the bone around the knee.

For many years, the primary cause for the development of osteoporosis after a spinal cord injury was attributed to disuse [125], potentially related in part to loss of the integrity of “muscle–bone” units [103]. However, that explanation cannot account for all of the bone loss as in space microgravity conditions; exercise can maintain muscle integrity but not that of bone (discussed in [1,5]). Thus, in space flight microgravity conditions there is an apparent uncoupling of such “muscle–bone units” where the bone loss continues even when muscles are activated and should still be releasing relevant molecules that affect bone. However, many aspects of this paradigm still remain to be better understood via more research.

The loss of bone after a SCI is complicated in females as such injuries can also affect menstrual cycles [126,127], and some aspects of bone turnover are altered during the menstrual cycle [25]. Furthermore, some bone markers, but not all, exhibit sex differences in spinal cord-injured individuals [128]. Therefore, bone loss in females with a SCI is potentially multifactorial.

Individuals with SCI are at risk for fractures [129,130,131,132]. Furthermore, those incurring fractures are also at risk for altered outcomes, such as non-unions [133,134]. Interestingly, Wang and colleagues [135] have reported that in men with SCI fractures have elevated serum leptin levels and lower callus formation than those with fractures and no SCI. As both individuals with and without a SCI and a fracture likely have their fractured limb immobilized in a disuse state, such findings as mentioned above indicate that factors other than those strictly related to disuse are at play in patients with a SCI, and possibly, this may relate to loss of neural regulatory influences.

This conclusion also has some support from preclinical models of fracture healing. Thus, in mice and rats, neuronal influences on fracture healing have been reported [136,137], including in SCI mice [138]. Thus, loss of neuronal input into a limb or fracture site can lead to compromised healing. While both human patients and rodent models indicate neuronal influences in fracture healing, the situation is complex due to the combination of bone loss, disuse, and loss of functional innervation.

## 4. Possible Role of Neural Input in Fracture Healing and Altered Healing with Brain Trauma

Acute bone loss occurs following a fracture (reviewed in [139]). This may be due to the immobilization of the fracture site with loss of biomechanical stimulation and thus loss of osteokines and muscle cytokines, as well as loss of blood supply and neural input at the site of injury (reviewed in [139]). In addition, there is also likely loss of neural regulation at the site of injury when the bone fractures.

Interestingly, it has been noted for decades [140,141,142,143] that trauma patients with a fracture and a brain injury appear to exhibit enhanced fracture healing [144,145,146,147]. However, not all of the evidence supports this contention (reviewed in [148]). Some variation in the link could potentially relate to host variables [149], including the age and sex of the patient, the location and extent of the brain injury/trauma, as well as any co-morbidities (i.e., diabetes) the patients may also have. Many of the observations cited regarding the potential link between fracture healing and brain trauma have focused on the timing and extent of callus formation.

A variety of humoral factors have been reported to be potentially responsible for the reported accelerated healing of fractures when accompanied by brain injury/trauma. These include prolactin [146,150], basic fibroblast growth factor (bFGF) [151,152], calcitonin gene-related peptide (CGRP) [153], nerve growth factor (NGF) and vascular endothelial growth factor (VEGF) [154], corticosteroids [155], inflammatory mediators [146], or other currently undefined humoral factors [156].

While the mechanisms responsible for the potential link between fracture healing and brain trauma in patients is not well defined, the observations have led to a number of preclinical studies designed to gain further insights into the phenomenon. In a rabbit model, both NGF and CGRP were implicated in fracture healing with concomitant traumatic brain injury [157] as well as the use of leptin in rabbit models [158]. However, most of the studies in preclinical models have focused on using rodent models [159,160,161]. Interestingly, in contrast to a single injury, repeated brain injury led to impaired fracture healing in male mice [162], potentially implicating inflammatory processes in the observations. In female rats, nerve-related genes have been detected in the fracture callus, but their detection decreases with age of the animals at the time of experimental fracture [163]. Whether the influence of brain injury/trauma on fracture healing in human patients was affected by the age of the patients was not found in the literature.

A considerable body of research has focused on the possible central role of leptin in the healing response following a fracture in brain-injured rodents [164,165,166]. This has also been shown in rodent models using leptin-deficient ob/ob mice [167,168]. However, in patients, it has been reported that leptin levels are low in patients with a long-bone fracture and a concomitant brain injury [169]. Such differences may be due to species differences in leptin effects, such as those noted in other conditions (discussed in [170]). Thus, the findings regarding leptin in rodent models may not translate well to the human condition.

Relevant to the above discussion are reports indicating that patients suffering a brain injury/trauma are at risk to develop heterotopic ossification [171,172,173,174]. Thus, this risk could be manifested at a site of a fracture in a patient suffering a brain injury/trauma. Furthermore, this risk for heterotopic ossification of soft tissues could be due to soluble mediators or perhaps via local neural effects from the tissue innervation. Of note, it has been discussed by O’Brien et al. [175] that surgical tendon trauma can also lead to heterotopic ossification. Furthermore, this has been shown in a mouse model that injury to one Achilles tendon leads to heterotopic ossification of the contralateral tendon, potentially through signals from the surgical tendon being transmitted to the contralateral tendon via innervation and the dorsal root ganglion [176]. Thus, the link between apparent accelerated fracture healing and brain injury/trauma is in need of additional mechanistic research to explore the role of humoral factors and the local release of neural factors in patient populations. A search of the literature did not uncover any reports of any contralateral effects following a lower limb fracture in combination with a brain injury.

## 5. Potential Neural Influences on Development of Post-Menopausal Osteoporosis (OP) and Age-Related OP

Bone appears to be regulated differently in females compared to males, likely due to variables associated with reproduction. For instance, there are alterations in bone turnover during the menstrual cycle in females [25]; there is bone loss during pregnancy and lactation [27,28,29] due to the transfer of needed calcium to the neonate to facilitate growth of the skeleton. In rare cases, overt osteoporosis can develop during pregnancy but resolves during the post-partum period [177]. After pregnancy or lactation, the bone loss can be reversed (discussed in [29,30]). Thus, females have the ability to regulate bone loss and recovery during this phase of the life spectrum prior to menopause.

While some authors have attributed such changes to changes in sex hormone levels, such as estrogen (discussed in [31]), perhaps it is not the only option, or the effect of the hormones is indirectly influencing bone regulation. As reviewed recently by Zhang et al. [178], there may also be a role for sympathetic nerves in osteoporosis. Sex hormones can influence a number of tissues, including neural tissues across the lifespan [178,179,180,181,182,183,184,185]. Sex hormones can also influence the neuromuscular function in muscle [186,187], and therefore, sex hormones could potentially influence nerves in bone resulting in altered regulation. That is not to say that sex hormones do not exert direct effects on muscle [188,189] and bone cells [190,191]. Thus, regulation of bone by sex hormones could be at multiple levels and indirectly via innervation and via direct effects of sex hormones on target tissues.

Furthermore, it is also possible that the direct effects of sex hormones could be primarily on catabolic regulation of bone turnover, while the neural influence could be mainly on bone cells contributing to bone synthesis. If such a dual system of regulation was operational, one could induce dysregulation of bone via elevating bone loss by enhancing bone catabolism and not altering bone synthesis or by not changing bone catabolism but stopping or slowing down bone synthesis. This is not a new concept in that Albright et al. [192] also suggested that post-menopausal OP was a disorder of bone formation (discussed in [193]) in spite of the fact that it is currently treated with drugs, such as bisphosphonates, which are aimed at inhibiting bone catabolism and resorption (discussed in [34]). Such a network of regulation could potentially explain the enhanced bone turnover during pregnancy and lactation versus the restoration of bone integrity in the post-lactation time frame. Interestingly, mice lacking the neurotransmitter substance P (SP) exhibit normal bone modeling when young but diminished bone formation and increased bone resorption with aging [77], which is somewhat consistent with a compromised neural system as described above. However, these mouse findings may indicate that there are additional regulatory factors operative when undergoing growth and maturation (i.e., anabolic factors) that decline with age allowing for the SP-deficiency to become evident. In addition, these mice still retained CGRP, which also can exert effects on bone [77]. A limitation of this report is that the authors only used male mice for some experiments or did not report the sex of the animals used, so it is not known if any sex differences in the impact of the SP-deficiency was evident. Whether this might be a rodent-specific situation is also currently unknown.

Of note, several reports from Seeman (discussed in [194,195,196,197]) have also stressed the potential importance of the periosteum, an innervated tissue surrounding bone [198], in the regulation of bone. Thus, bone deposition and removal appear to have site-specific regulatory elements. Such different influences could play varying importance across the lifespan and between the sexes.

However, it is still not clear how such a multicomponent system would lead to only a subset of females developing OP following menopause and how OP develops in an even smaller subset of males. The development of age-related OP in males may be better explained by an age-related loss of nerve–bone regulation, as it is known that there is loss of innervation of many tissues with age [199,200,201,202,203,204]. In humans, loss of innervation of bone may be influenced by individual factors, such as genetics and epigenetics, putting some individuals at a higher risk for developing OP than others, irrespective of sex.

While the above discussion regarding a role for nerves in the development of OP in post-menopausal females and a subset of males is somewhat speculative at this juncture, the concept has some support conceptually and is certainly amenable to future research. In fact, development of OP in post-menopausal females could be only one aspect of the potential post-menopausal conditions that may have a basis in dysregulation of the neural-tissue interface for a variety of conditions that development following menopause (discussed in [31]).

## 6. Space Flight, Bedrest, and Neural Regulation

### 6.1. Influence of Space Flight and Bedrest on Bone Regulation

As stated previously, bone as well as muscles and other MSK tissues appear to operate under the “use it or lose it” paradigm. One way to lose bone is via disuse, such as staying in bed and not engaging GRFs [1].

Disuse by removing oneself from GRFs, such as being immobilized/sedentary for the majority of the day for many days in a row can lead to the rapid loss of bone as an adult, occurring within days [205]. This occurs via uncoupling of bone maintenance with the excess of bone resorption, but the rate of bone loss is quite variable between individuals (reviewed in [6]), again possibly indicating that multiple steps are involved beyond loss of stimulation via GRF exposure.

While prolonged bedrest can result from diseases or loss of mobility via aging and frailty, bed rest with six-degree head-down tilt is also used as a terrestrial surrogate for space flight (reviewed in [1,5]). For the latter, the participants are usually healthy younger males and females, although a current study is comprised of older individuals more reflective of astronaut ages or even older. This bedrest surrogate also allows for testing of potential countermeasures, such as exercise, short-arm centrifuges, or even drugs, such as bisphosphonates [206] and other anti-resorptive drugs that could be used in space (reviewed in [207,208]). One caveat of such bedrest studies is that as they are performed on Earth, the loss of bone still is occurring in the presence of 1 g gravity but without the GRF loading.

In contrast, bone loss during space flight or living at low Earth orbit conditions on a vehicle, such as the International Space Station (ISS), occurs in microgravity and the absence of GRF loading. Similar to bone loss during bedrest, the rate of bone loss in astronauts is quite variable (discussed in [6]), indicating astronauts are heterogeneous with regard to their response to microgravity conditions. Furthermore, astronauts lose bone mostly in the lower extremities; the extremities are subjected to more GRF loading than the upper extremities. As evolution could not have anticipated space flight and exposure to microgravity, the finding of a heterogenous response to microgravity regarding bone loss implies that such variation arose in the systems controlling the rate of bone loss when they would be silent while on Earth, or potentially contributing to bone loss during aging. This is in contrast to the post-menopausal development of osteoporosis in a subset of females where bone loss is associated with loss of hormonal influence. Furthermore, as most astronauts have been males, but aged males with osteoporosis represent only 25% of patients with OP, there are some differences between astronauts losing bone in space and males on Earth developing OP on Earth. Thus, there are levels of complexity regarding the regulation of bone integrity that remain to be answered.

### 6.2. Is There a Role for Neural Regulation Dysfunction during Bone Loss during Space Flight and Osteoporosis?

It is clear from the above discussion that bone is innervated, loss of innervation from SCI or stroke leads to loss of bone, bone is regulated differently in females and males, and bone loss can occur in response to a number of conditions, including space flight and the Earth analogue, prolonged bedrest. For many of these altered environments, the bone loss has been mainly attributed to the loss of mechanical loading, with counter measures focused on the use of exercise to overcome the “disuse atrophy”. However, a role for the innervation of bone in the loss of bone density with space flight and prolonged bedrest has not been explored in detail by the biomechanists, who have focused more on the direct effects of loading on bone cells rather than indirectly via the potential regulatory functions of mechanical loading on neural elements that in turn could influence bone cells. Furthermore, insights into the heterogeneity in bone loss associated with microgravity conditions has not been explained.

Similarly, the focus of research on bone loss associated with menopause has been directed at the direct effects of hormones on bone cells in spite of the fact that only a subset of females develops clinically relevant osteoporosis after menopause. And similar to the heterogeneity of bone loss in microgravity, why females with osteoporosis also exhibit a variation in the rate and extent of bone loss has not been explained. Thus, in both of these situations, microgravity and post-menopausal osteoporosis, several aspects of the bone loss remain unaccounted for at the present time.

With regard to hormonal regulation of bone, as discussed earlier, it is clear that bone cells can be affected by hormones, particularly in females. Most cells in the body express both nuclear and plasma membrane receptors for estrogen and progesterone, as well as androgens [209,210,211,212,213,214], with the non-genomic plasma membrane receptors mediating rapid responses. Hormones, such as estrogens, can influence a variety of neural activities [215,216,217,218,219], so it is not outside the realm of possibility that such hormones may influence the functioning of neural activity in bone. In a rat model, the influence of neuronal signals was diminished in estrogen-deficient females [220], potentially indicating a loss of neural regulatory mechanisms after menopause could contribute to bone loss. Depending on the extent of loss of the neural regulation (local synthesis versus dependence on systemic levels of hormones), there could be variation in the impact of the disruption of the neural influence. This variation could contribute to the development of osteoporosis and the rate of bone loss in the post-menopausal environment, but this will require more focused research to address this possibility. However, it should be pointed out that the regulation of bone is complex, particularly in females where the regulation must involve the integration of a variety of factors (mechanical loading, sex hormones, and innervation/nerves) that can vary across the lifespan. Thus, integration and establishment of which mechanisms have priority depends in part on whether one is pre-puberty, skeletally mature but cycling or pregnant/lactating, or in the post-menopausal state. The other variable that may play a role in the nature of the integration of factors regulating bone integrity is that of epigenetic modification [221], modifications that can occur at the life transition points or due to life experiences. Thus, epigenetic modifications, dependent in part on which cells are affected and where they are located, could alter the regulation of bone by any of the known variables (mechanical loading, neural input, hormonal factors) at different stages of life. However, this area will require additional research effort to better understand the local versus systemic impact of epigenetic alterations on bone regulation.

Interestingly, most of the bone loss in microgravity by astronauts has been in males, so the complexities of deciphering the underlying neural regulatory mechanisms may be less than in females. However, as more female astronauts go into space for longer periods of time, the findings with the two sexes may be compared even though the numbers of each is still fairly small. As mentioned earlier, male astronauts lose bone in microgravity at variable rates, and exercise in microgravity conditions is only partially effective. This set of circumstances likely cannot be explained solely by a loss of mechanical loading as the only factor involved; although, it is possible that the exercises used in space do not adequately reflect loading on Earth. An additional potential explanation is that bone is regulated by a combination of factors, including neuroregulatory factors, and the conditions in space lead to loss of regulation by multiple factors and the exercise conditions in space do not replicate the conditions required to restore the neuro-component. Such a conclusion would also be supported by the effect of exercise on bone health in patients with spinal cord injuries, where such exercise is again only partially effective, indicating factors in addition to mechanical loading are required for a complete pattern for maintaining bone integrity. In this circumstance, it is clear that the neuro-component has been damaged and its contributions disrupted by the injury. Interestingly, the fact that bisphosphonates can alleviate bone loss both in space and after a SCI likely indicates that directly influencing bone–cell activity can block or interfere with the impact of loss of effectiveness of the regulatory systems.

Bone regulation in post-puberty females is more complex than for males given that hormonal variation during the menstrual cycle, pregnancy, and lactation can also lead to modifications of bone integrity, modifications that are apparently mostly reversible. While bone cells express sex hormone receptors and thus could be directly influenced by hormones, it is also possible that some of the effects of sex hormones on bone cells is indirect via modulation of the regulatory activity of the neural elements. Whether these neural elements affected by sex hormones are those directing elements innervating bones or more central control elements in the brain or via the dorsal root ganglion would remain to be determined.

A role for a central neural (i.e., brain)-localized mechanism may also be hypothesized to explain some of the variation in bone loss accompanying space flight and menopause. As only a subset of females develops clinically relevant post-menopausal osteoporosis and there is variation in the rate and extent of bone loss, variants of a central mechanism might better explain the systemic effects of osteoporosis rather than local effects in a variety of mechanical environments, local environments that may be subjected to individual epigenetic modifications [31,222]. Similarly, as space flight could not have been anticipated by evolution, the finding that male astronauts experience variable bone loss during space flight may mean that variation in a central regulatory mechanism (i.e., neural) may be silent during most of the lifespan while being maintained in a 1 g environment on Earth. However, in some males, there may be age-related loss of regulation at the neural level, leading to clinical osteoporosis, but at numbers much less than for post-menopausal females. Such heterogeneity between individual humans may reside in genetic variation that may not be evident if the individual remains on Earth.

In this model of bone regulation with contributions from neural elements (likely central), mechanical loading, and sex hormones (mainly from females), the various regulatory systems must be integrated to allow for optimal functioning or functioning in a manner that allows for the regulation of bone within a window of physiological intactness throughout most of the lifespan. There may be some loss of integration with aging, and there must be some plasticity to achieve priority setting during development and then followed by growth and maturation. In this model of integration, some regulatory systems not only interact with bone cells directly but also interact with other regulatory systems to affect bone indirectly (i.e., sex hormones such as estrogen affecting neural regulation or ground reaction force the loading affecting both bone cells and the neural regulatory system). Some of these potential regulatory options are depicted in Figure 1.

Regarding the options depicted in Figure 1, they are only examples of possible forms of regulation by these three elements. There would likely be differences between the sexes with regard to how they functioned, which bones in different locations in the body that may be regulated in an environment-specific manner by these examples (i.e., lower extremities, upper extremities, long bones versus those of the hand), and the relationships between different regulatory variables may vary across the lifespan in an age, sex, and race manner [163,223,224]. Thus, for females, the critical importance of the hormones may vary with puberty, the menstrual cycle, pregnancy and lactation, and menopause given the evolutionary priority of reproduction. This priority may indicate that this hormonal aspect of bone regulation may, at times, be of the highest influence. In contrast, the hormonal contribution to bone regulation in males may be primarily focused on growth and maturation, puberty, and andropause/aging. Likely, there may not be an a priori reason to expect that one set of regulatory factors would be constant for both sexes across the lifespan based on the previous discussion in earlier sections of this review.

The concept that neural regulation plays a role in the regulation of bone is not unique to this tissue. For instance, SCI can affect not only bone regulation but also several other organs that would not be influenced by the disuse or lack of mechanical loading (reviewed in [225]). Other organs affected, depending on the level of the injury, include the bladder, lungs, spleen, kidneys, and the pineal gland. The findings with the spleen discussed by Wulf and Tom [225] is interesting as like bone, tissues of the immune system are disseminated throughout the body, operate in different environments, and are innervated [226,227,228,229]. The spleen [230,231], the lymph nodes [232], the gut mucosal immune system [233,234], the thymus [235], and the bone marrow [236] are all innervated, and the innervation plays a regulatory role. Innervation of tissues, such as the spleen, exhibit some species-specific innervation differences [231], and in rodents, the integrity of some immune system components, but not all, can decline with age [235]. Thus, there may be some species-specific aspects to the regulatory control of the immune system, as well as potentially bone. In addition, some aspects of bone regulation may be altered during the aging process as alluded to earlier. Importantly, there may be lessons learned from regulatory control by neural elements of tissues other than bone that could be applied to the study of bone going forward. In addition, there may be some human-specific and sex-specific aspects to the inter-relationships between mechanical loading, sex hormones, and neural regulation, as female humans uniquely have menstrual cycles and experience menopause, and the human brain may have capabilities related to cognition that other species do not possess.

This topic area should be the focus of research going forward as it may lead to both new understanding of bone regulation and also the development of new treatment options for the loss of bone integrity. Included in such studies should be the study of astronauts as space flight offers a unique set of conditions that not only could not have been anticipated by evolution but also provides an environment relatively free of disease in the traditional sense. Additionally, further study of the enhanced fracture healing associated with head/brain trauma may provide some further insights into the signals and neuromediators contributing to bone regulation [237], possibly associated with neuroinflammation in the brain leading to enhanced release at the site of fractures.

## 7. Conclusions

Considerable research over the past decades has focused on the role of mechanical loading on bone growth, homeostasis, and loss of integrity. While many insights regarding bone regulation have been revealed, an in-depth study of other regulatory variables (i.e., neuroregulation and hormonal regulation) and their integration into the dynamics of bone regulation in different locations of the body at different times in the lifespan and between the sexes has not occurred with the development of a coherent perspective of the complexity of bone regulation. The advent of space flight and the exposure to microgravity has provided a unique set of insights into the complexity of bone regulation, and further investigation of regulation in this environment could provide further understanding of the interrelationship between neural and mechanical loading of bone in male and female astronauts. Regulation of bone in females across the lifespan is likely more complex than that in males, with differences associated with reproduction evident, and thus, hormonal influences may play a prominent role in bone regulation in females while on Earth. In females, the integration of the direct effects of hormones on bone cells and the indirect effects via hormone effects on neuroregulatory elements provides additional complexity that needs further investigation. Furthermore, as females are heterogeneous, and not all experience post-menopausal OP, a role for genetics and epigenetics must also be factored into regulation. In addition, genetics variables may also play a role in males as some males can develop age-dependent OP. Thus, from the discussion presented, regulation of bone is complex and dynamic across the lifespan and involves variables beyond mechanical loading in different locations and environments, and thus, the boundary conditions of earth (i.e., 1 g, geomagnetic field, temperature range) and the biological features of *Homo sapiens* have shaped both homeostasis and responses to perturbations of these tissues. As outlined in Figure 1, the relative contributions of mechanical loading, hormones, and neural contributions, as well as genetic and epigenetic factors, are potentially very complex, and likely one set of circumstances or relationships does not fit all. Therefore, future research may need to focus on bone regulation in individual *Homo sapiens* rather than populations to gain more detail in this regard.

## Figures and Tables

**Figure 1 biomolecules-13-01136-f001:**
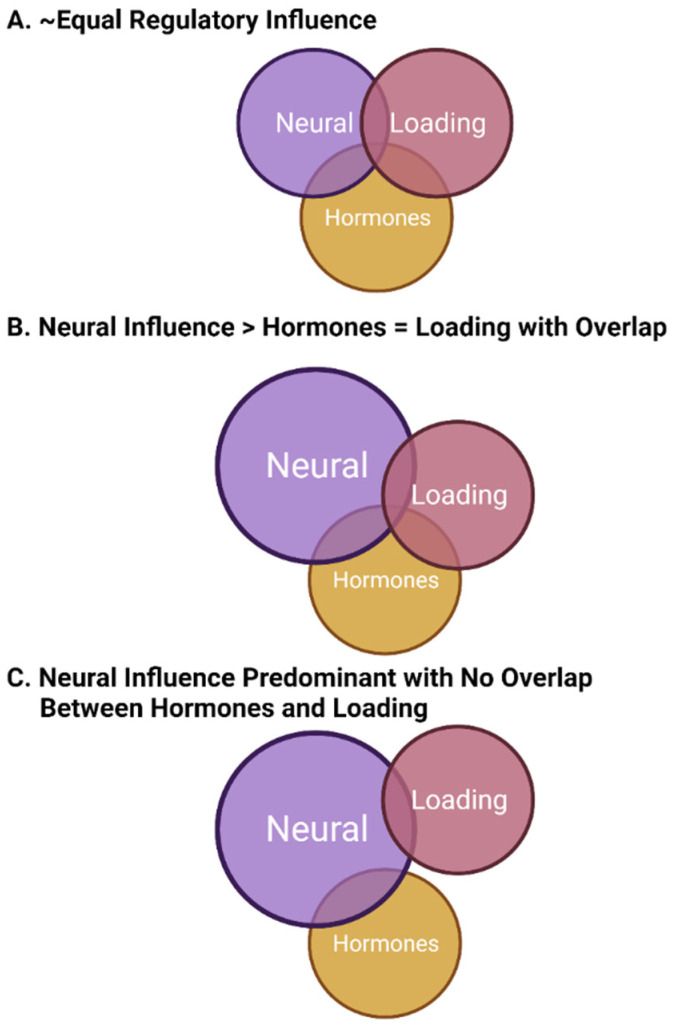
Potential regulatory relationships between mechanical loading, sex hormones, and neural elements in the regulation of bone. The three panels (**A**–**C**) represent examples of such relationships where the size of the circle and the overlap between circles indicate variation in regulatory influence, as well as potential integration. Such relationships may vary between the sexes, within the sexes (particularly in females and possibly due to genetic heterogeneity in humans), and at different time points during the life cycle, so there may not be a single set of relationships for bone in different environments and locations. This figure was prepared in BioRender.

## Data Availability

Not applicable.

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
