# Peer review of "Regulation of Bone by Mechanical Loading, Sex Hormones, and Nerves: Integration of Such Regulatory Complexity and Implications for Bone Loss during Space Flight and Post-Menopausal Osteoporosis"

_biomolecules, 2023, doi:10.3390/biom13071136_

Round 1
Reviewer 1 Report
The paper presents a review of bone remodeling under specific conditions such as osteoporosis and space flight (micro-gravity) based on sex, hormones and nerves. The paper is clearly written, clearly presented and clearly understandable. Most developments are well argued and propose some new insight to the problem of bone remodeling.
However, a few points need to be addressed as they are not completely defendable in the proposed context and some shortcuts need to be nuanced. See below.
1- The different points presented below are all focus on one main assumption that will be discussed lower down.
Point 1 : paragraph 2 line 203 to 208 “such a system ...on the ISS”
Point 2 : paragraph 6 line 488 to 492 “However, ...influence bone cells”
Point 3 : paragraph 6 line 536 to 539 “A more plausible … neuro-component”
Point 4 : paragraph 6 line 557 to 559 “A prominent … menopause”
Point 5 : Figure 1
Point 6 : paragraph 6 line 603 to 604 “The concept …. this tissue”
Although it is clearly written almost everywhere in the text that neural influence on bone regulation is only one parameter and may play a role, when reading the paper, it is felt that the author clearly thinks that it plays definitely a major role and that bone remodeling has little functional interpretation without this effect.
Bone remodeling, as the author states, is a very complex mechanism and many factors influence its evolution. It is also accepted that most probably neural effects play a role in this evolution. However, there is no evidence anywhere of the quantification of this effect (in percent compared to the many other effects), nor its variability as a function of the external environment.
It is also agreed that the sole mechanics is not enough to understand bone remodeling and that weight (in terms of gravity, muscles contraction or osteoporosis resorption) are not enough to understand it (we could mention for example a few examples such as biology, cells, biochemistry, genetics, etc ... which are all patient dependent). However, the sole addition of the neural effect, although it may play a role, is certainly not enough either.
We encourage the author to revise the overall text of the paper accounting for this aspect as a function of the overall environment for which neural aspects is one of the possible parameter among many others that impact bone remodeling.
2- It seems there is not paragraph 7
3- Conclusion – line 644-645 “Clearly … prominent” : As far as the text of the paper goes, there is no proof of this affirmation and again, it is only one factor among many others, even accounting with the previous sentence, this is hard to argue. Yes male and female have different bone regulations, but again many factors are influenced and most probably the hormones are one of the driving parameter.
Reviewer 2 Report
This review aims at discussing the relative contribution of various regulatory components, namely mechanical loading, sex hormones and neural elements, for bone integrity. The paper is well organized and clearly written. However, the following issue should be addressed:
The author only mentions the Piezo channels as mechanosensitive structures sensing mechanical strain in osteocytes (line 176). However, other structures including the PTH receptor type 1 (JBMR 30:1231-44, 2015; J Cell Physiol 237:3927-43, 2022) also seem to be involved in this respect (reviewed in Histo Histopathol 32:751-60, 2017). This should be considered and mentioned in the corresponding paragraph.
Other minor concerns:
Several acronyms across the text should be defined: MSK (line 124), ISS (line 208, NPY, CGRP (line 236).
Some mispellings must be amended: CGRP0 (line 347); "Tthere" (line 590)
Round 2
Reviewer 1 Report
The author has adequately answered the comments made in the first review. There is no need for extra correction.